# Entropy Measures of Electroencephalograms towards the Diagnosis of Psychogenic Non-Epileptic Seizures

**DOI:** 10.3390/e24101348

**Published:** 2022-09-23

**Authors:** Chloe Hinchliffe, Mahinda Yogarajah, Samia Elkommos, Hongying Tang, Daniel Abasolo

**Affiliations:** 1Centre for Biomedical Engineering, School of Mechanical Engineering Sciences, University of Surrey, Guildford GU2 7XH, UK; 2Department of Clinical and Experimental Epilepsy, Institute of Neurology, University College London, National Hospital for Neurology and Neurosurgery, University College London Hospitals, Epilepsy Society, London WC1E 6BT, UK; 3Neurosciences Research Centre, St George’s University of London, London SW17 0RE, UK; 4Atkinson Morley Regional Neuroscience Centre, St George’s Hospital, London SW17 0QT, UK; 5School of Neuroscience, Institute of Psychiatry, Psychology and Neuroscience, King’s College London, London WC2R 2LS, UK; 6Department of Computer Science, University of Surrey, Guildford GU2 7XH, UK

**Keywords:** psychogenic non-epileptic seizures, machine learning, entropy

## Abstract

Psychogenic non-epileptic seizures (PNES) may resemble epileptic seizures but are not caused by epileptic activity. However, the analysis of electroencephalogram (EEG) signals with entropy algorithms could help identify patterns that differentiate PNES and epilepsy. Furthermore, the use of machine learning could reduce the current diagnosis costs by automating classification. The current study extracted the approximate sample, spectral, singular value decomposition, and Renyi entropies from interictal EEGs and electrocardiograms (ECG)s of 48 PNES and 29 epilepsy subjects in the broad, delta, theta, alpha, beta, and gamma frequency bands. Each feature-band pair was classified by a support vector machine (SVM), k-nearest neighbour (kNN), random forest (RF), and gradient boosting machine (GBM). In most cases, the broad band returned higher accuracy, gamma returned the lowest, and combining the six bands together improved classifier performance. The Renyi entropy was the best feature and returned high accuracy in every band. The highest balanced accuracy, 95.03%, was obtained by the kNN with Renyi entropy and combining all bands except broad. This analysis showed that entropy measures can differentiate between interictal PNES and epilepsy with high accuracy, and improved performances indicate that combining bands is an effective improvement for diagnosing PNES from EEGs and ECGs.

## 1. Introduction

Psychogenic non-epileptic seizures (PNES) clinically resemble epileptic seizures but are not due to epileptic electrical brain activity [1]. Although the condition is almost as prevalent as multiple sclerosis [2,3], PNES is regularly misdiagnosed: people with PNES are not appropriately diagnosed for an average of seven years [4], and approximately 78% of patients were taking at least one anti-epileptic drug at the time of accurate diagnosis [5]. This has serious adverse effects for both patients and healthcare systems, through unnecessary visits to hospitals, medical tests, and treatments. In addition, since anti-epileptic drugs are not effective for PNES, these misdiagnosed patients will have endured the negative side effects of these expensive drugs without any significant benefit [3]. Furthermore, an estimated one in five referrals to epilepsy clinics actually have PNES [6], highlighting the difficulties in making an accurate diagnosis.

The current gold standard method of diagnosis is the recording of a seizure with video-electroencephalogram (EEG), from which a specialist assesses the semiology (the clinically observable features of the seizure) and visually inspects the EEG [7]. While this method is reliable, there are several shortcomings. Not all epileptic seizures are associated with qualitatively identifiable ictal EEG abnormalities [8], and EEG has a relatively poor ability to accurately identify a patient without epilepsy, with a sensitivity of 25–56% [9]. As a result, it can sometimes be difficult to differentiate between epileptic and psychogenic seizures. Therefore, given the need for in-patient admission and prolonged video-EEG recording, this diagnostic method is costly, inconvenient for the patient, and not accessible to all hospitals [3].

Entropy is a measure of the randomness and uncertainty of a signal, and higher entropy indicates a more complex or chaotic system [10]. It has a low computation cost and has been shown to be effective by previous researchers, making it a suitable option for machine learning. Entropy measures of the EEGs of PNES subjects have been of interest to previous researchers. Pyrzowski et al. [11] used interval analysis of interictal EEGs to compare 51 epilepsy subjects to 14 PNES and 14 headache (PNES and epilepsy free) subjects. The EEGs were theta-alpha filtered and the zero crossing rates were histogram pooled and normalised across the segments and/or channels. From this, the relative counts of fixed-length intervals, several statistical measures, and the Shannon and minimum entropies were extracted. The researchers found that only the entropy measure significantly separated the epilepsy and non-epilepsy (headache patients included) without being affected by the presence of antiepileptic drugs. The researchers found that Shannon entropy was the better of the two entropies at separating epilepsy and non-epilepsy. Furthermore, for Shannon entropy, the optimum frequency band was 7–13 Hz and performed best for temporal-occipital channels, specifically T6 + T5.

One study [12] used 314 epileptic, non-epileptic, and healthy subjects. From these EEGs, 14 EEG features including spectral entropy were extracted using the empirical wavelet transform (EWT), singular spectrum empirical mode decomposition (SSEMD), and singular spectrum empirical wavelet transform (SSEWT). The researchers compared plots of the different types of seizures (e.g., focal non-epileptic, complex partial, interictal non-epileptic, normal controls, etc.) for these features and with each extraction method. For the EWT and SSEMD, the spectral entropies of the epileptic and non-epileptic groups overlap. The SSEWT method, however, shows that spectral entropy has some separation, with the normal subjects, focal non-epileptic, and tonic-clonic seizures isolated from the other classes. Complex partial and generalized non-epileptic seizures are also separated from the other seizure types but overlap each other.

Gasparini et al. [13] and Lo Giudice et al. [14] both used the entropy of the EEG as a control for comparison to the entropies of hidden layers in deep learning models. Gasparini et al. [13] extracted the Shannon and permutation entropies from the EEGs of six PNES subjects and ten healthy controls and found no statistical difference between the two classes for either measure. Lo Giudice et al. [14] used interictal EEG from 18 epilepsy and 18 PNES subjects and also found no statistical difference between the two classes for the permutation entropy of the signal.

ECG analysis has also been a focus of previous PNES research. Ponnusamy, Marques, and Reuber in 2011 [15] and 2012 [16], and Romigi et al. [17] all extracted the approximate entropy (ApEn) from the heart rate (RR interval data) as a part of extensive heart rate variability (HRV) analysis. Ponnusamy, Marques, and Reuber’s 2011 [15] study found that, interictally, ApEn was significantly lower in PNES than in healthy controls, but there was no statistical difference in interictal HRV ApEn between PNES and epilepsy subjects. Their 2012 study [16] found no statistical difference in ictal ApEn between PNES and epilepsy groups. However, epileptic subjects showed a decrease in ApEn during seizure activity, whereas PNES subjects did not. Romigi et al. [17], however, found that ApEn decreased in PNES subjects during seizure activity compared to at rest, before, and after the attack. Furthermore, there was no difference between subjects with PNES only and PNES subjects with comorbid epilepsy.

Single biomedical signal parameters have been shown to be insufficient as a differentiator for PNES and epilepsy [18]. Therefore, a potential tool to mitigate these problems is machine learning. Machine learning classifiers are mathematical algorithms that “learn” how to separate conditions by training on a set of data. The validity of this trained model is then tested using more data. When analysing biomedical signals, these data are typically comprised of one or more features extracted from the signal taken at different observations. This allows the classifiers to consider multiple factors with different types of information simultaneously. The model’s ability to separate the conditions is assessed using performance metrics such as accuracy (ability to predict both conditions correctly) [19].

Machine learning has been previously used to classify entropy measures extracted from the EEGs of PNES patients. For instance, from 2014–2018, a series of six papers published by the same group of researchers [20,21,22,23,24,25] used spectral entropy as one of 55 EEG features analysed by machine learning.

Ahmadi et al. [26] used EEGs from 20 epilepsy and 20 PNES subjects and compared the Shannon entropy, spectral entropy, Renyi entropy, Higuchi fractal dimension, Katz fractal dimension, and the EEG frequency bands with an imperialist competitive algorithm. They found that spectral entropy and Renyi entropy were the most important EEG features as they were always among the five best feature subsets. Furthermore, the classification accuracy decreased significantly when either or both were excluded from a subset. They also found that SVMs with a linear or RBF kernel were the best classifiers.

The same group did another study [10], this time with five epilepsy and five PNES subjects. They extracted the same EEG features from each frequency band, this time including the energy of the signal. The researchers found that beta was the best band for all features and gamma was the worst. The highest performing features differ for each band, making an overarching conclusion difficult.

Cura et al. [27] used synchrosqueezing to represent the time-frequency maps of 16 epilepsy and six PNES subjects. From these maps, 17 features were extracted: three flux, flatness, and energy concentration measures; two Renyi entropy measures; six statistical features; and five TF sub-band energy measures. The researchers used decision tree, SVM, RF, and RUSBoost classifiers to differentiate all 17 features. For the three class problems, the inter-PNES (non-seizure), PNES seizure, and epileptic seizure EEGs, the highest accuracy, precision, and lowest false discovery rate were reported by RF with 95.8%, 91.4%, and 8.6%. The highest sensitivity was reported by the RUSBoost classifier with 90.3%. All classifiers except the SVM reported higher accuracy ≥ 93%, sensitivity ≥ 82%, and precision ≥ 86% and lower false discovery rates ≤ 14% values. The researchers also compared the inter-PNES and PNES EEGs for PNES seizure detection. All accuracies were ≥90% (excluding the SVM for one patient) and RF reported the highest of these.

This paper will aim to assess the ability of seven entropy metrics to differentially diagnose PNES and epilepsy by using these features individually as the inputs for four popular machine learning methods. This analysis will compare the diagnostic power of each feature and each EEG frequency band for a large database of PNES and epilepsy EEG and electrocardiogram (ECG) recordings.

## 2. Materials and Methods

The data used in this analysis were collected routinely at St George’s Hospital, London and consisted of interictal and preictal surface EEG recordings from 48 PNES and 29 epilepsy patients. The PNES subjects have an age range of 17–59 (mean 34.76 ± 10.55) and a male/female ratio of 14/34. The epilepsy subjects have an age range of 19–79 (mean 38.95 ± 13.93) and a male/female ratio of 18/11. Suitable cases were retrospectively identified from the video-EEG database of those attending for inpatient video-EEG monitoring from 2016 to 2019. The diagnosis of functional seizures was made according to International League Against Epilepsy diagnostic criteria [28] by at least two clinicians experienced in the diagnosis of epilepsy and were documented through video-EEG in all cases. The diagnosis of epileptic seizures was based upon EEG confirmed ictal epileptiform activity during the recorded epileptic event during video-EEG monitoring. Exclusion criteria for both groups included cases with a dual diagnosis of both epileptic and functional non-epileptic seizures. The recordings were taken with Natus Networks with an EEG32 headbox. The EEG electrodes were placed according to the 10–20 system montage with Cz-Pz as the reference electrode. The ECG is comprised of two electrodes, ECG+ and ECG-, placed on the right and left mid-clavicular line. The sampling frequencies were either 256, 512, or 1024 Hz, and bandpass filtering from 0.5 to 70 Hz was applied. The data were reviewed and clipped by experienced clinicians in the field, who selected awake time epochs when patients were still and at rest, without seizures or ictal/epileptiform manifestations, and with minimal noise. All clipped EEG data was de-identified and the video removed prior to the current analysis. Anonymised recordings were stored in EDF+ format.

The EEGs and ECGs were preprocessed using MNE-python [29]. The signals with a sampling rate of over 256 Hz were downsampled to this value and the common electrodes were selected: Fp1, F7, T3, T5, O1, F3, C3, P3, Fz, Cz, Fp2, F8, T4, T6, O2, F4, C4, P4, Fpz, Pz, ECG+, and ECG-. The EEGs were filtered using an FIR, Hamming window, bandpass filter with cutoff frequencies of 0.5 and 40 Hz. The ECGs were filtered using a Bessel IIR bandpass filter with cutoff frequencies of 0.25 and 40 Hz, the method for which was derived from [30,31]. Inspection of the time and frequency plots of the EEG showed no significant mains noise, so this was not specifically removed. The data were then segmented into ten-second non-overlapping epochs. To remove noise, epochs where the EEG amplitude did not exceed 1 µV were removed, and AutoReject [32] automatically removed epochs with noisy EEG. The remaining epochs were then visually inspected to exclude any epochs that contained flat EEG or ECG. The resulting 10,452 epochs were then baseline corrected using the average of each subject’s EEG. These EEG samples were then filtered into the frequency bands: delta 0.5–4 Hz, theta 4–8 Hz, alpha 8–13 Hz, beta 13–30 Hz, and gamma 30–40 Hz. The ECG channel was found by subtracting the values of the ECG+ lead from the ECG- lead. Baseline wander was then removed using a filter with a 0.05 Hz cutoff [33]. Entropy features were extracted from every band and every channel (including ECG), including the original broad band (0.5–40 Hz). The ECG filtering, however, was the same for each EEG frequency band analysed (0.25–40 Hz).

The entropy measures used in this analysis were: approximate, sample, spectral, singular value decomposition (SVD), Renyi, and wavelet entropy. These features were extracted from each channel in each sample, giving 21 input parameters per band per feature. The approximate and sample entropies were computed using EntropyHub [34], the spectral and SVD entropies were calculated using MNE-features [35], and the Renyi entropy was estimated using DIT [36].

Approximate entropy was introduced by Pincus [37] to define irregularity in sequences and time series data [38]. Formally, given N data points from a time series $\left\{x\left(n\right)\right\}=x\left(1\right),\;x\left(2\right),\ldots ,\;x\left(N\right)$, the ApEn is calculated using two input parameters, a run length m and a tolerance window r, which must be fixed [38]. To define $\mathrm{ApEn}\left(m,r,N\right),$ form vector-sequences $X\left(1\right),\ldots ,X\left(N-m+1\right)$ defined by $X\left(i\right)=\left[x\left(i\right),\;x\left(i+1\right),\ldots ,\;x\left(i+m-1\right)\right]$, where i=1,…, N−m+1. Then define the distance $d\left[X\left(i\right),X\left(j\right)\right]$ between vectors $X\left(i\right)$ and $X\left(j\right)$ as the maximum distance in their respective scalar components. For each i≤N−m+1, construct $C_{i}^{m}\left(r\right)$ defined as (the number of $X\left(j\right)$ such that $d\left[X\left(i\right),X\left(j\right)\right]\leq r)/\left(N-m+1\right)$. Next, define $\Phi^{m}\left(r\right)$ as the average value of $\ln C_{i}^{m}\left(r\right)$. The ApEn is then defined in Equation (1) [38], where N is 2560 throughout this analysis.
(1)$$\mathrm{ApEn}\left(m,r,N\right)=\Phi^{m}\left(r\right)-\Phi^{m+1}\left(r\right)$$

Nevertheless, to avoid the occurrence of ln(0) in the calculation of ApEn, the algorithm includes self-matching, leading to a discussion of bias in this entropy metric [39]. Sample entropy (SampEn) was introduced by Richman and Moorman [39] as an improvement upon ApEn by reducing the dependency on record length and to avoid self-matching. To define $\mathrm{SampEn}\left(m,r,N\right)$ of a time series $\left\{x\left(n\right)\right\}=x\left(1\right),\;x\left(2\right),\ldots ,\;x\left(N\right)$, with a run length m and a tolerance window r, form vector-sequences $X_{m}\left(1\right),\ldots ,X_{m}\left(N-m+1\right)$, defined by $X_{m}\left(i\right)=\left[x\left(i\right),\;x\left(i+1\right),\ldots ,\;x\left(i+m-1\right)\right]$, where i=1,…,N−m+1. The distance $d\left[X_{m}\left(i\right),X_{m}\left(j\right)\right]$ between vectors $X_{m}\left(i\right)$ and $X_{m}\left(j\right)$ is then defined as the maximum absolute distance between their respective scalar components. For each i≤N−m, construct $B_{i}^{m}\left(r\right)$ defined as (the number of $X_{m}\left(j\right)$ such that $d\left[X_{m}\left(i\right),X_{m}\left(j\right)\right]\leq r)/\left(N-m-1\right)$. Next, define $B^{m}\left(r\right)$ as the average value of $B_{i}^{m}\left(r\right)$. Then, increase the dimension to m+1 and calculate $A_{i}$ as the number of $X_{m+1}\left(i\right)$ within r of $X_{m+1}\left(j\right)$, where j ranges from 1 to $N-m\left(j\neq i\right)$. Define $A_{i}^{m}\left(r\right)$ as $A_{i}/\left(N-m-1\right)$ and $A^{m}\left(r\right)$ as the average value of $A_{i}^{m}\left(r\right)$. Therefore, $B^{m}\left(r\right)$ is the probability that two sequences will match m points, whereas $A^{m}\left(r\right)$ is the probability that two sequences will match m+1 points. Sample entropy is then defined using Equation (2),
(2)$$\mathrm{SampEn}\left(m,r\right)=\underset{N\to \infty }{\lim}\left\{-\ln\left[\frac{A^{m}\left(r\right)}{B^{m}\left(r\right)}\right]\right\}$$
which is estimated by the statistic in Equation (3), where N is 2560 throughout this analysis.
(3)$$\mathrm{SampEn}\left(m,r,N\right)=-\ln\left[\frac{A^{m}\left(r\right)}{B^{m}\left(r\right)}\right]$$

Since both ApEn and SampEn are highly dependent on the input parameters run length m and tolerance window r, these values require selection. For both entropies, the recommended range of values for the parameters are m=1 or 2 and r between 0.1 and 0.25 times the standard deviation (SD) of the input time series $x\left(n\right)$ [39]. Therefore, the following parameter combinations were tested with a grid search $m=\left[1,2\right],\;r_{SD}=\left[0.1,\;0.15,\;0.2,\;0.25\right]$, where $r=r_{sd}\times$ is the SD of the input time series. To avoid overfitting the data, a subset of ten patients per class were selected for this analysis. ApEn and SampEn were extracted from this subset using each combination of m and r. These features were then inputted to a support vector machine (SVM) with a radial basis function (RBF) kernel and validated with 5-fold cross validation. The m and r combination that returned the highest average balanced accuracy from the classifiers was then selected as the input parameters to be used for the analysis with the full dataset. The specifics of the machine learning aspects of this process are described below.

Spectral entropy (SpecEn) finds the Shannon entropy [40] of the power spectrum and is calculated using Equation (4), where $p_{i}$ is the probability distribution of the power spectrum of the time series, i is one of the discrete states (assuming a bin width of one spectral unit), the sum of $p_{i}$ is 1, and Ω is the number of discrete states [41].
(4)$$\mathrm{SpecEn}\left(f\right)=-\frac{1}{\ln\left(\Omega \right)}\overset{\Omega }{\underset{i=1}{\sum }}p_{i}\ln\left(p_{i}\right)\;$$

SVD entropy (SVDEn) was defined by Alter et al. [42]. SVD is a matrix orthogonalisation decomposition method, so for a time series $\left\{x\left(n\right)\right\}=x\left(1\right),\;x\left(2\right),\ldots ,\;x\left(N\right)$ the Hankel matrix $H_{m\times n}$ can be reconstructed as
(5)$$H_{m\times n}=\left[\begin{array}{cccc} x\left(1\right) & x\left(2\right) & \cdots & x\left(n\right)\\ x\left(2\right) & x\left(3\right) & \ldots & x\left(n+1\right)\\ \vdots & \vdots & \ddots & \vdots \\ x\left(m\right) & x\left(m+1\right) & \cdots & x\left(N\right) \end{array}\right]$$
where 1<n<N, m=N−n+1 [43]. The SVD of $H_{m\times n}$ can be defined as
(6)$$H_{m\times n}=U\sum V^{T}=\left[u_{1},u_{2},\;\ldots ,\;u_{L}\right]\left[\begin{array}{cccc} \sigma_{1} & 0 & \cdots & 0\\ 0 & \sigma_{2} & \ldots & 0\\ \vdots & \vdots & \ddots & \vdots \\ 0 & 0 & \cdots & \sigma_{L} \end{array}\right]\left[\begin{array}{c} v_{1}\\ v_{2}\\ v_{L} \end{array}\right]$$
where the left singular vectors $U_{m\times m}$ and right singular vectors $V_{n\times n}$ are orthogonal matrices, and $\sum_{m\times n}$ is a diagonal matrix composed of singular values $\left(\sigma_{1}\geq \sigma_{2}\geq \ldots \geq \sigma_{L}\geq 0,\;L=\min\left(m,n\right)\right)$ [43]. In this space, matrix $H_{m\times n}$ satisfies $\langle k|\sum |l\rangle \equiv \sum_{l}\delta_{kl}\geq 0$ for all 1≤k, l≤L [42]. Let us define the normalised eigenvalues as,
(7)$$p_{l}=\sigma_{l}{}^{2}/\overset{L}{\underset{k}{\sum }}\sigma_{k}{}^{2}$$
which indicates the relative significance of the lth eigenvalue and eigenvector in terms of the fraction of the overall expression that they capture [42]. Then the SVD entropy of the dataset X is as shown in Equation (8) [42]:(8)$$\mathrm{SVDEn}=-\frac{1}{\log\left(L\right)}\overset{L}{\underset{k=1}{\sum }}p_{k}\log_{2}\left(p_{k}\right)$$

Renyi entropy (REn) estimates the spectral complexity of a signal and is calculated using Equation (9), where the order  α≥0 and α≠1, $p_{i}^{\alpha }$ is the probability distribution of the time series, i is one of the discrete states, and Ω is the number of discrete states [44]. For this analysis, α=2 to replicate [10] for ease of comparison with this study.
(9)$$\mathrm{REn}\left(\alpha \right)=\frac{1}{1-\alpha }\log_{2}\left(\overset{\Omega }{\underset{i=1}{\sum }}p_{i}^{\alpha }\right)$$

Wavelet entropy (WaveEn) is a measure of the degree of disorder associated with the multi-frequency signal response. The wavelet coefficients $C_{i,j}$ were found using wavelet decomposition, where i is the time index and j is the index of the different resolution levels. The energy for each time i and level j can be found using Equation (10) [45].
(10)$$E_{i,j}=C_{i,j}{}^{2}$$
The mean energy was then calculated using Equation (11),
(11)$$E_{j}^{\left(k\right)}=\frac{1}{n}\overset{k_{0}+∆t}{\underset{i=k_{0}}{\sum }}E_{i,j}$$
where the index k is the mean value in successive time windows, which will now give the time evolution; $k_{0}$ is the starting value of the time window $\left(k_{0}=1,\;1+∆t,\;1+∆t,\;\ldots \right)$; and n is the number of wavelet coefficients in the time window for each resolution level [45]. The probability distribution for each level can be defined using Equation (12) [45].
(12)$$p_{j}^{\left(k\right)}=\frac{E_{j}^{\left(k\right)}}{E_{tot}^{\left(k\right)}}$$ Following the definition of Shannon entropy [40], the time-varying wavelet entropy was found using Equation (13) [45]. More details can be found at [46].
(13)$${\mathrm{WaveEn}}^{\left(k\right)}=-\underset{j}{\sum }p_{j}^{\left(k\right)}\ln p_{j}^{\left(k\right)}$$ For this analysis, Morlet wavelets were used since they are commonly used in EEG research [47].

Once these features had been extracted from every channel for every epoch in every band, they were used to train and test four machine learning classifiers: SVM, k-nearest neighbours (kNN), random forest (RF), and gradient boosting machine (GBM). These models were implemented using the scikit-learn python package [48].

SVMs were introduced in [49] and classify by searching for an optimal hyperplane that separates the classes. If the data are separable, the hyperplane maximises a margin around itself that does not contain any data, creating boundaries for the classes. Otherwise, the algorithm establishes a penalty on the length of the margin for every observation that is on the wrong side. The SVM classifiers used in this analysis used an RBF kernel, which maps the data onto a non-linear plane. The RBF kernel between two patterns x and x’ is calculated using Equation (14).
(14)$$K\left(x,x’\right)=exp\left(-\gamma ||x-x’{||}^{2}\right)$$ In this case, γ was taken as 1/(number of features × variance of the data).

The kNN algorithm is based on the idea that similar groups will cluster. The model is trained by ‘plotting’ observations based on their features, presumably with the classes clustering. The algorithm is tested by plotting an observation and classifying it based on the class of the nearest neighbours. The number of nearest neighbours, k, was individually selected by a grid search that tested 2, 3, 4, 5, 6, 7, 8, 9, 10, 12, 15, and 20 neighbours. This defined k as the value that returned the highest balanced accuracy with ten-fold cross validation.

RF was introduced by [50] and is based on randomised decision trees. Decision trees are flowchart-like structures that predict the value of a target variable by learning a series of simple decision rules based on the training data. RF uses an ensemble of trees, each with a different random subset of the features in a method called bootstrap aggregating, or bagging. This decreases the variance, compared to an individual decision tree, and reduces the risk of overfitting. The class was then taken as the average of the trees’ probabilistic predictions, whereas the original publication [50] let each tree vote for a single class.

GBMs are ensembles of weak learners, typically decision trees, and were introduced by [51,52]. GBMs are similar to gradient descents in a functional space. The model is built by adding a new tree with every iteration. The new tree is fitted to minimise the sum of the losses of the (now previous) model. For binary classification, a prediction is made based on the probability that the sample belongs to the positive class. This is found by applying the sigmoid function to the tree ensemble.

To classify the feature set, ten-fold cross validation was used to define the training and testing datasets. Since the classes in this dataset are imbalanced with more PNES data, the epilepsy data in the training set was oversampled using a synthetic minority over-sampling technique (SMOTE). The feature space was then reduced using principal component analysis (PCA), with a variance of 95%.

Precision, recall, and balanced accuracy were used to evaluate the classifiers’ predictions of test data. Since the dataset was imbalanced, these metrics were selected as they avoid inflated performance metrics on imbalanced datasets. Equations (15)–(17) show the calculations for these performance metrics.
(15)$$\mathrm{precision}=\frac{TP}{TP+FP}$$
(16)$$\mathrm{recall}=\frac{TP}{TP+FN}$$
(17)$$\mathrm{balanced}\;\mathrm{accuracy}=\frac{1}{2}\left(\frac{TP}{TP+FN}+\frac{TN}{TN+FP}\right)$$
where *TP* is the true positive rate, *TN* is the true negative rate, *FP* is the false positive rate, and *FN* is the false negative rate. Here, PNES is the positive class and epilepsy is the negative class.

Permutation feature importance was also used to compare the EEG frequency bands. This was done by adapting the algorithm [50] to include multiple features. A model m was fitted using training data, and then a reference score s was defined using the validation data D. Each feature (channel) of the set (band) to be assessed $f_{n:o}$ was then permutated (randomly shuffled) in order to corrupt the validation samples of that band and give ${\tilde{D}}_{k,n:o}$. The score ${\tilde{s}}_{k,n:o}$ of model m on this corrupted validation dataset was then computed. This process of permutating and calculating score ${\tilde{s}}_{k,n:o}$ was repeated K times with iteration k. The importance $i_{n:o}$ of the feature set (band) $f_{n:o}$ is then defined using Equation (18).
(18)$$i_{n:o}=s-\frac{1}{K}\overset{K}{\underset{k=1}{\sum }}s_{n:o}$$

## 3. Results

The grid search to establish the ideal values for m and $r_{SD}$ found that the highest average accuracy across the bands was returned when m=2 and $r_{SD}=0.2$ for ApEn and when m=1 and $r_{SD}=0.15$ for SampEn. These parameters were then used to extract the ApEn and SampEn from the full dataset. The accuracies from these tests can be found in the Appendix A.

Using the methods described, the balanced accuracies returned are reported in Table 1. Tables containing the precision and recall can be found in the Appendix A.

Table 1 shows a range of balanced accuracies with only two instances returning below chance (50%). The highest accuracy was 94.68%, with 96.12% precision and 95.19% recall, which was obtained by Renyi entropy with a kNN classifier in the ‘all’ band. Generally, the lowest performing entropy measure was wavelet entropy, and the best was Renyi entropy. Overall, the lowest accuracies were obtained by the gamma band, and with all the EEG bands combined—the ‘all’ band—the highest accuracies were returned.

When comparing the entropy measures and the frequency bands, it is possible to group the measures into three different trends: Renyi entropy; sample, approximate, SVD, and spectral entropy; and wavelet entropy. Wavelet entropy was the measure returning the lowest accuracies with a mean of 53.24 ± 3.18%. This measure returned higher accuracies in the ‘all’ and theta bands, and the lowest accuracies in the alpha, beta, and gamma bands.

Sample, approximate, SVD, and spectral entropy returned higher accuracies in the ‘all’ and broad bands. The combined ‘all’ band improved the SVM, kNN, and GBM classifiers. The RF, however, only showed a slight increase. The SVM accuracy was significantly improved (over 12% increase, excluding spectral entropy) by the ‘all’ band for all these measures, as well as the kNN (over 9% increase, excluding spectral entropy). The delta, theta, alpha, and beta bands returned medium accuracies, and the gamma band returned a further drop in classifier performance. These measures typically outperformed wavelet entropy by a large margin, with means of 67.71 ± 7.29%, 68.97 ± 8.13%, 65.23 ± 7.44%, and 65.16 ± 5.93%, respectively.

The Renyi entropy was overall the highest performing entropy measure, with a mean of 82.48 ± 4.20%. In the broad band, the accuracies of this measure were only somewhat higher than the sample, approximate SVD, and spectral entropies. However, the accuracies for Renyi entropy increased in the theta, alpha, beta, and gamma bands. In comparison, the accuracy for the other measures remained stable or decreased in these bands, especially gamma. The combination of ‘all’ bands improved the accuracy, especially for the SVM, which increased by 10.86%. As a result, most of the classifiers in the ‘all’ band were able to achieve over 90%.

The best classifiers were kNN and, generally, the higher the overall accuracy for a band and/or feature, the bigger the difference between kNN and RF and the other two classifiers. Overall, RF was the better classifier. However, the kNN returned the highest accuracy value since it, along with the SVM, was greatly improved by combining all the bands, whereas RF and GBM were less affected. Furthermore, Table 1 shows that GBM was often the lowest performing classifier.

Since the combination of the bands performed well, a further experiment was conducted to establish which specific bands were contributing to the high accuracy. Using the same process as described above, each band was excluded from the full set and the remaining bands were used for classification. The ECG signal was also used as an input for each band. This experiment used the highest performing classifier, kNN, and the highest performing entropy metric, Renyi entropy, and the outcomes are summarised in Table 2. The importance of the band reported is the average permutation band importance over ten-fold cross validation.

Table 2 shows that removing a single band had a minor effect on the precision and recall, thus affecting the balanced accuracy but not significantly. Excluding broad and delta increased the accuracy to 95.03% and 94.93%, respectively, from 94.68% when all bands were used. However, excluding the others resulted in a loss of 0.60% or more. Therefore, the theta, alpha, beta, and gamma bands contain important information for Renyi entropy. The band importance from the permutation-based testing is congruent with these findings, with the broad and delta bands returning half the permutation importance of the other bands. These findings are congruent with the trend shown in Table 1 for the Renyi entropy, where broad and delta slightly underperformed compared to the other four non-combination bands.

## 4. Discussion

Spectral and wavelet entropy were both found by calculating the Shannon entropy of the frequency spectrum, where spectral entropy estimated the spectrum using Welch’s method and wavelet entropy used Morlet wavelets. Despite these similarities, the resultant accuracies were significantly different, with spectral entropy outperforming wavelet entropy in every band and with every classifier. This suggests that Welch’s method is more suitable for extracting the uncertainty in the frequency domain for this specific task. Furthermore, the spectral and wavelet entropies both returned the lowest accuracies, on average, of all the measures. Therefore, our results suggest that for these data measures of complexity, those in the time domain may be more effective than those in the frequency domain. The measure that returned the highest accuracy, Renyi entropy, is a variation of Shannon entropy applied directly to the time series. This further lends to the effectiveness of temporal complexity, and further research should explore similar methods.

While the classifier performances for most entropy measures were improved by combining all frequency bands, generally the SVM and kNN improved more significantly than the decision tree-based algorithms, especially RF. Decision trees do not need to increase the parameters with more inputs, so it is possible that the extra information was lost for these model types. Furthermore, the nature of an ensemble of random subsamples of the feature set, as is the case with RF, may have hindered the classifier’s ability to consider the extra information. This could be the cause of the limited improvement and occasional degradation of the RF when combining the classifiers, despite the high performance in the non-combination bands. Therefore, feature selection methods, such as feature ranking, should be used with this classifier to potentially improve accuracy with larger feature sets.

A 2021 meta-analysis on resting state EEGs for the diagnosis of epilepsy and PNES [53] found that comparing oscillations along the theta band may separate epilepsy and PNES. Reuber et al. [4] also found interictal slow rhythms in the theta band for nine out of 50 PNES patients. When considering only the delta, theta, alpha, beta, and gamma bands, the current analysis found that the theta band returned the highest balanced accuracy for 13 out of 24 (four classifiers for six entropy measures) instances, indicating that a difference in theta oscillations could be reflected in the entropy. However, the beta band returned the highest accuracy in 8 of these 24 instances, especially for the spectral entropy. Therefore, the beta band could also be of interest to future researchers.

Comparison to the literature is complex due to the difference in techniques used to analyse the EEGs of PNES patients. For instance, Pyrzowski et al [11] extracted the entropy from pooled histograms of the zero crossing rate, and the six-paper series [20,21,22,23,24,25] and Cura et al. [27] only used one or two entropy measures as part of a larger feature set, obscuring the influence of the entropy. Furthermore, [11,20,21,22,23,24,25] included non-PNES subjects within their subject cohorts. The papers that included the ECG [15,16,17], all analysed the entropy of the heart rate data, a binary signal representing the R peaks, instead of the ECG signal itself. While these studies do represent the potential of entropy for this diagnostic task, the fundamental difference in method makes comparisons with them impossible.

Gasparini et al. [13] and Lo Giudice et al. [14] both statistically analysed the entropy of the EEG signal. The authors of [13] found no differences between the Shannon or permutation entropies of PNES patients and healthy controls, and [14] found no difference in interictal permutation entropy between PNES and epilepsy subjects. Therefore, statistical analysis alone may not be sufficient to differentiate between these groups.

The studies published by Ahmadi et al. [10,26] give details of the performance of similar entropy measures and classifiers in the frequency bands and use PNES-only and epilepsy-only groups. Thus, an in-depth comparison with the current study is possible, although neither study used an ECG channel, only EEGs, and only include the interictal state. The 2018 study [26] used an imperial competitive algorithm to rank the individual feature-band pairs and has listed the top five combinations of inputs for each classifier. They found that RF and decision trees were the weaker classifiers, compared to SVM-Linear, SVM-RBF, and GBM. However, the current analysis found that RF was overall the best classifier, with GBM underperforming. Ahmadi et al. (2018) also found that spectral and Renyi entropies were the most important features, compared to Shannon entropy, Higuchi fractal dimension, and Katz fractal dimension. The current study did not extract Shannon entropy or any fractal dimensions, so a direct comparison cannot be made. However, this analysis did find that Renyi entropy was a very high-performing metric for all bands, and spectral entropy was better than chance (50% accuracy) for all tests. Ahmadi et al. (2018) do not directly compare the frequency bands, though gamma is not listed in the features for any of the top performing inputs. This is congruent to the current study, since gamma underperformed for most entropy measures, including spectral entropy. The outlier is Renyi entropy, which retained high accuracies in the gamma band in the current analysis. Furthermore, broad band Renyi entropy was listed by [26] for most of the top performing combinations. By comparison, the current study found that Renyi was the entropy measure that returned the highest accuracies for the broad band analysis but returned lower accuracies than the other bands for this metric. In addition, the delta band is not noted as important by [26] for either entropy measure; therefore, it was found to be less important for these features, which is in agreement with the findings of the current analysis.

The study by Ahmadi et al. 2020 [10] gave a clearer breakdown of the bands for the Shannon, spectral, and Renyi entropy, although only the precision and recall values were reported, not accuracy, and the broad band was not analysed. In addition, the values reported for the delta and theta bands are exactly the same, which is statistically unlikely and is not reflected in the ROC curves also given. Therefore, the values reported in the current version of this paper for one of these bands may be incorrect. The delta, theta, and gamma bands for all entropy measures and Shannon entropy in the alpha band all return low performance metrics of roughly chance accuracy. The beta band, and spectral and Renyi entropy in the alpha band, however, return mostly 70% precision and 60% recall. ROC analysis showed that the beta band outperformed the delta, theta, alpha, and gamma bands. The alpha band performed well, but much worse than the beta. The delta and theta bands were similar to random chance, and gamma distinctly underperformed for all measures. For the current analysis, the Renyi entropy does show that delta is one of the bands less likely to help differentiate PNES from epilepsy, but disagrees for the theta, alpha, beta, and gamma bands, which all return good and fairly similar accuracies. These trends reported by Ahmadi et al. (2020) were more similar to those for the sample, approximate, spectral, and SVD entropies; where gamma significantly underperformed. Spectral entropy also showed a slight increase in beta band accuracies, but only spectral entropy showed this, and delta performed on par with the other bands.

A limitation of our study is that the two classes are not age- or sex-matched. The ages are similar enough that significant influence is unlikely. However, the PNES group has significantly more females than males, whereas the epilepsy group has more males than females. This is due to PNES being more commonly diagnosed in females than males by a factor of 3:1 [54,55]. In previous studies [56,57,58], machine learning has been successfully used to separate EEG entropy measures of females and males; therefore, it is possible that the balanced accuracies were inflated by the disparity in sex between the two groups. To ensure that this disparity did not have a significant impact, the model that returned the highest accuracy (Renyi entropy with a kNN classifier, with the delta, theta, alpha, beta, and gamma bands inputted as separate features) was trained and tested again with a subset of subjects that were age- and sex-matched. This matched dataset included 50 subjects with a ratio of 11 females to 14 males in both classes, and the epilepsy group had a mean age of 39.16 ± 11.86 while the PNES group had 38.52 ± 10.96. The accuracy, precision, and recall of the matched dataset were 95.40%, 97.10%, and 93.33%. Therefore, the balanced accuracy and precision increased slightly while the recall decreased slightly. Considering this outcome and the similarities in the literature, it is still reasonable to conclude that the difference in sexes between the classes had a minor impact and that entropy measures are indeed powerful measures in differentially diagnosing PNES and epilepsy. Another limitation is that the data includes both preictal (before seizure) and interictal (resting) recordings. Therefore, it is not possible to separate the impacts of these different types of data on the results. Finally, due to a small patient cohort, the current study used ten-fold cross-validation to assess the classifiers. Therefore, samples from each subject were present in both the training and testing datasets. While this is a limitation, it demonstrates that this method is viable and, if trained on a larger population, could be beneficial in clinical contexts.

## 5. Conclusions

This study shows that the analysis of different frequency bands in the EEG, plus the ECG, with different entropy algorithms returns useful information for the classification of PNES. Furthermore, the bands providing the highest accuracy vary from entropy measure to measure. Therefore, the combination of bands for classification by machine learning algorithms can return higher results. While this would increase the computation cost, entropy measures are quick and low-cost; therefore, the added computation is a small cost compared to the improved performance. The current analysis found that the highest balanced accuracy, 95.03%, was returned by the delta, theta, alpha, beta, and gamma bands combined for the Renyi entropy when a kNN was used in the classification. However, this high performance may have been affected by the use of epoch-wise ten-fold cross validation. The kNN and RF classifiers returned the overall highest accuracies, with the GBM repeatedly underperforming compared to the others, and SVM and kNN showed more improvement with the combination of the bands. Further analysis should explore the combination of further low-cost features to increase the performance and improve the robustness of the classifiers for different patients.

## Figures and Tables

**Table 1 entropy-24-01348-t001:** Balanced accuracies of the entropy metrics for every classifier and EEG frequency band (ECG is included in every band). Bold values denote the highest accuracy amongst the classifiers for each EEG band and entropy measure.

Features	Classifiers	All	Broad	Delta	Theta	Alpha	Beta	Gamma
Renyi entropy	SVM	91.41%	75.95%	74.74%	80.55%	80.17%	79.36%	78.14%
	kNN	**94.68%**	83.17%	80.23%	87.73%	**88.29%**	**89.41%**	**87.83%**
	RF	92.75%	**83.29%**	**81.13%**	**88.11%**	87.38%	89.17%	87.18%
	GBM	81.63%	71.27%	71.97%	76.57%	76.65%	74.36%	76.26%
Sample entropy m=1, r=0.15*SD	SVM	84.11%	71.55%	66.67%	67.09%	**63.83%**	63.05%	58.58%
	kNN	**86.64%**	77.61%	64.61%	66.49%	60.29%	65.85%	59.87%
	RF	79.92%	**77.76%**	**67.96%**	**69.70%**	62.75%	**66.46%**	**62.00%**
	GBM	73.62%	67.48%	63.59%	65.13%	61.21%	62.10%	59.99%
Approximate entropy m=2, r=0.2*SD	SVM	85.66%	73.04%	64.97%	67.79%	67.59%	62.20%	58.50%
	kNN	**87.82%**	78.17%	64.07%	68.28%	67.13%	68.23%	60.87%
	RF	80.87%	**78.70%**	**67.02%**	**72.22%**	**68.52%**	**68.34%**	**63.06%**
	GBM	74.18%	68.60%	63.30%	65.54%	62.95%	62.58%	60.83%
SVD entropy	SVM	**83.26%**	69.28%	62.44%	64.16%	62.81%	64.28%	**56.58%**
	kNN	82.37%	72.22%	59.49%	62.48%	61.14%	61.82%	53.30%
	RF	76.72%	**74.84%**	**64.55%**	**66.16%**	**63.80%**	**65.82%**	55.88%
	GBM	72.29%	66.81%	61.78%	63.10%	60.00%	63.11%	55.87%
Spectral entropy	SVM	**79.03%**	69.24%	62.84%	62.98%	63.01%	65.16%	56.25%
	kNN	77.34%	72.92%	61.15%	60.40%	62.65%	65.20%	54.10%
	RF	72.24%	**74.79%**	**65.35%**	**63.62%**	**65.17%**	**68.36%**	**58.67%**
	GBM	69.44%	67.04%	61.80%	61.92%	60.71%	64.76%	58.32%
Wavelet entropy	SVM	**58.30%**	**54.72%**	**54.22%**	**60.62%**	50.88%	**50.94%**	50.19%
	kNN	53.57%	52.24%	52.48%	56.87%	50.95%	49.89%	49.54%
	RF	55.26%	53.23%	52.43%	58.96%	50.35%	50.73%	50.60%
	GBM	57.05%	52.36%	52.14%	59.01%	**51.07%**	50.72%	**51.48%**

**Table 2 entropy-24-01348-t002:** Precision, recall, and balanced accuracy of the kNN classifier trained and tested on Renyi entropy for all EEG frequency bands, excluding the corresponding band. ‘None’ denotes all bands are included with no exclusions. Band Importance shows the premutation importance of the band. The ECG channel was included in all iterations.

Band Excluded	Precision	Recall	Accuracy	Band Importance
Broad	96.40%	95.48%	95.03%	0.052
Delta	96.18%	95.63%	94.93%	0.062
Theta	95.90%	94.27%	94.08%	0.111
Alpha	95.65%	94.59%	94.03%	0.132
Beta	95.73%	94.54%	94.07%	0.128
Gamma	95.64%	94.57%	94.01%	0.114
None	96.12%	95.19%	94.68%	-

## Data Availability

The data used in this study were provided by St George’s Hospital and are not publicly available.

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
