# Peer review of "Entropy Measures of Electroencephalograms towards the Diagnosis of Psychogenic Non-Epileptic Seizures"

_entropy, 2022, doi:10.3390/e24101348_

Round 1

Reviewer 1 Report

Please see attached report.

Reviewer 2 Report

The review of the existing literature considering the entropy in EEG analysis is nicely presented. The subject of the paper is quite interesting. The methods, however, give an impression that the authors have implemented ready-made functions they are not familiar with. 

 The well-known, extensively used, and also critically analyzed Approximate entropy is incorrectly presented. The authors start with “First, a delayed reconstruction ?1:? for ? data points with embedding dimension ? and lag ?.” referencing the classical paper of Pincus [32]. The authors did not explain what m and ? are. In addition, the lag ? is not introduced by Pincus, but by other authors in another paper. If the lag is used, it should be explained what it is and proper reference where it was introduced should be quoted. Further on, Equation (1) is incorrect: namely, in quoted [32] that introduces ApEn, and generally, in any implementation of ApEn, the self-matching is not excluded due to the possibility of a logarithm of zero.

 ApEn is a parametric method. Numerous papers consider the minimum length of the time series N for reliable results, as well as the corresponding parameters m and R (? is usually set to one default value, find the literature). Poor choice of parameters gives inconsistent and erroneous estimates of entropy. Thus, the choice of parameters should be discussed, the original paper [32] was written long ago and the method has been critically revised. It can be concluded (but without certainty) that the entropy was estimated from a time series of length N = 2560 (10 s k 256 samples / s), but this is only an assumption and may not be correct.

 If students were to do this homework assignment, they would inevitably have to check the entropy for several parameters in order to choose the right ones. In this paper, which is intended for the journal, the authors ignore the parameters and do not consider it important to say which ones they used and what is the basis of their choice.

 Then equation (3) also contains an error, N=m+1 should be N-m+1.

 The authors consider ?1:? as data points, not as vectors of specific length as is the usual interpretation. But, introducing SampEn, they change the terminology “within range points“ from ApEn into “vector pairs”, Number of such pairs is A and B, but the author did not define what are these vectors, and what makes “a pair” of them. Indeed, the authors need to define the numbers A and B using a similar formula as in Equation (1), and bearing in mind that Richman and Moorman [33] excluded self-matching (there are many papers on self-matching and why SampEn excludes it). A relation between Ni, A, and B should be given: from the current explanation it seems that ApEn and SampEn have no relationship at all, while they are quite similar and should be explained in the same manner.

 The explanation of ApEn and SampEn given by the authors leaves the impression that they are either trying to confuse the reader or are far from knowing entropy and information theory.

 Other entropies are defined in a similar confused manner. Reny entropy is given by Eq. (8) where ???(?) is stated to be “the probability distribution of a signal ? of length ?, and ? is one of the discrete states [36].” But, if Reny entropy is given by the formulation stated in equation (8), then x is a discrete random variable, and ???(?) are the probabilities of its outcomes. In this context, however, x is a time series of length N. It is discrete in the sense any signal acquired by A/D conversion is discrete (the authors did not state the type of A/D conversion), but the number of its amplitude levels is sufficiently large to be regarded as a continuous signal. Its estimated probability distribution function does not have N “discrete states”, but less.

Then the authors say “Log energy entropy is given by Equation 6”, and they are wrong because this entropy is given by Equation 9. Again, the summation is not over N pi(x) values. However, the Log energy entropy concept is suspicious, but it is the fault of reference 37, which the authors, unfortunately, uncritically quoted. Unlike the squared signal amplitudes, the probability squared has no relation to energy whatsoever. Regardless, the minus sign in any entropy expression is intended to make an entropy positive, as the logarithm of probability is less than zero. In Eq. (9) the logarithms are squared yielding a positive value. The sum of positive values is positive as well, so the minus sign is unnecessary unless we wish to have negative entropy.

 At this point, the review stops. The authors uncritically implemented ready-made software tools but their text does not prove that they are aware of the particularities of each method implemented. If they did not prove that the parameters for feature evaluation were properly chosen, then the discussion on the results is not reliable.

 Some minor technical comments, besides the ones already written: references are sometimes in square brackets, sometimes in parenthesis. Line 175 starts with a full stop. Sometimes a space is missing between the number and the unit (0.05Hz).

Round 2

Reviewer 2 Report

The authors have made a notable effort to improve their work. In the new version, they have addressed all the issues that had not been well described (if at all described) in their previous version. The paper is now much more substantial and also easier to follow. I am glad to say that I have no more comments, as the topic of the paper is quite interesting, and this is an important contribution. It only needed to be written in a way to get the approval of the broader audience and not to get the impression that it was written in haste.